# A broadly tunable synthesis of linear α-olefins

Andreas Gollwitzer [1], Thomas Dietel [1], Winfried P. Kretschmer[1] & Rhett Kempe [1]

The catalytic synthesis of linear α-olefins from ethylene is a technologically highly important reaction. A synthesis concept allowing the formation of selective products and various linear α-olefin product distributions with one catalyst system is highly desirable. Here, we describe a trimetallic catalyst system (Y–Al–Ni) consisting of a rare earth metal polymerization catalyst which can mediate coordinative chain transfer to triethylaluminum combined with a simultaneously operating nickel β-hydride elimination/transfer catalyst. This nickel catalyst displaces the grown alkyl chains forming linear α-olefins and recycles the aluminum-based chain transfer agent. With one catalyst system, we can synthesize product spectra ranging from selective 1-butene formation to α-olefin distributions centered at 850 gmol$^{-1}$ with a low polydispersity. The key to this highly flexible linear α-olefin synthesis is the easy tuning of the rates of the Y and Ni catalysis independently of each other. The reaction is substoichiometric or formally catalytic regarding the chain transfer agent.

[1] Lehrstuhl für Anorganische Chemie II—Katalysatordesign, Universitätsstr. 30, Universität Bayreuth, 95440 Bayreuth, Germany. Correspondence and requests for materials should be addressed to R.K. (email: kempe@uni-bayreuth.de)

The oligomerization of ethylene to linear α-olefins (LAOs) is an important industrial process with an increasing annual total world production of multiple megatons[1]. It is one of the most important technological applications of homogeneous catalysis and, thus, intensively investigated in industry and academia[2–5]. The LAOs produced range from 1-butene to various product distributions and are used to produce detergent alcohols, lubricants, plasticizers, fine and oil field chemicals, and as co-monomers for polyethylenes[2–5]. Highly flexible syntheses of LAOs with just one catalyst system would be highly desirable to match the frequently changing market demands. Coordinative chain-transfer polymerization (CCTP)[6–8], a polymerization protocol characterized by polymeryl chain-transfer between dormant states of a chain-transfer agent (CTA) and a polymerization catalyst responsible for the chain growth, allows the controlled polymerization of ethylene towards metal-terminated polyethylenes. A fundamental problem associated with CCTP is the inverse first-order dependence of the rate of the chain growth from the CTA concentration, which restricts the number of chains that can be grown efficiently by a catalyst molecule, also called (low) catalyst economy[9]. A solution to this problem would be a CCTP process that features a substoichiometric or formally catalytic use of the chain-transfer agent. Gibson and coworkers described the sequential combination of CCTP via chain growth at zinc and the subsequent displacement using a nickel catalyst[10]. In this two-step procedure, product formation is stoichiometric to the Zn alkyl. It was shown recently that the addition of [Ni(acac)₂] (acac-H=pentane-2,4-dione) to the Gibson CCTP catalyst, to run a tandem process, does not influence the CCTP process, as no displacement reaction was observed[11]. The addition of iron complexes, such as [(bipy)FeEt₂] (bipy=2,2′bi-pyridine, Et=ethyl), permits CTA recycling and result in a poisoning of the CCTP catalyst in combination with a very slow displacement reaction[11]. A slow displacement reaction reduced the flexibility of the synthesis drastically. We expected that by choosing the right combination of CCTP catalyst, displacement catalyst and CTA, a broadly tunable LAO synthesis process would become feasible which permits the synthesis of different olefin product distributions and selective LAO formation with one catalyst system. Recently, we introduced the corresponding concept, the combination of CCTP and alkyl chain displacement via β-H elimination/transfer for α-olefin synthesis[12].

Herein, we report on a highly flexible synthesis of LAO in which a trimetallic catalyst system permits the selective formation of an α-olefin, namely 1-butene, or of various LAO distributions. The catalyst system consists of an yttrium polymerization catalyst able to mediate controlled chain transfer towards triethylaluminum (TEA) and a Ni complex capable of controlled and efficient β-hydride elimination/transfer of aluminum alkyls of various chain lengths. The key to the broad product spectrum is the independent adjustability of the rates of the Y- and Ni-catalyzed subprocesses. In addition, the two catalysts do not poison each other significantly. Our LAO synthesis protocol is also substoichiometric or formally catalytic regarding Al. Thus, the problem of low catalyst economy or low CTA to catalyst ratios is addressed by efficient CTA recycling.

## Results

**The trimetallic LAO synthesis concept and catalysts mediating it**. The concept is schematically shown in Fig. 1a.

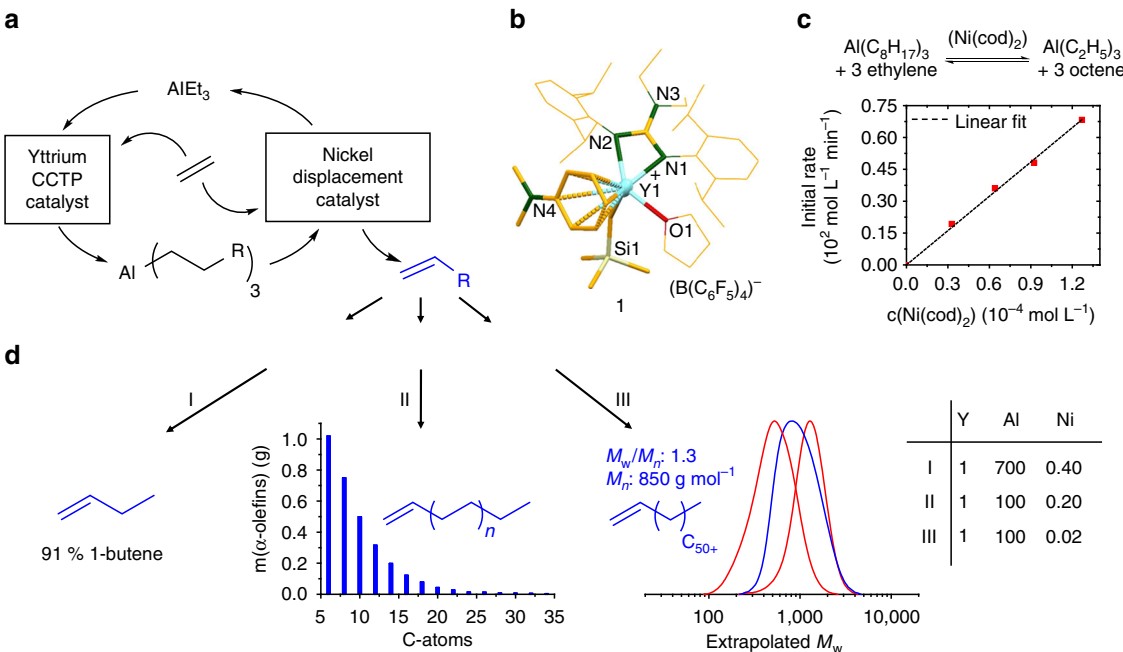

**Fig. 1** A broadly tunable synthesis of linear α-olefins applying a trimetallic catalyst system. **a** Combination of an yttrium coordinative chain transfer polymerization (CCTP) catalyst, triethylaluminum (TEA), and a nickel chain displacement catalyst (β-H elimination/transfer) permits the highly flexible formation of α-olefins via variation of the rates of the two catalytic steps. It is not urgent that all three ethyl groups on aluminum are exchanged; partial chain exchange to Al can be sufficient. **b** Molecular structure of the yttrium catalyst **1** determined by X-ray single crystal structure analysis. **c** Kinetic investigations (¹H NMR spectroscopy based) of the [Ni(cod)₂]-catalyzed octyl chain displacement reaction (β-H elimination/transfer) of Al alkyls. Plot of the initial rates (−dc(Al(C₈H₁₇)₃)/dt (10² mol⁻¹ l⁻¹ min⁻¹)) vs. different Ni catalyst concentrations (10⁻⁴ mol l⁻¹) indicating the reaction is first order regarding Ni (slope = 0.5385 ± 0.0080×10⁶ min⁻¹, linear fit: adj. R square = 0.9988). **d** Product scope. Variation of the concentrations of the catalysts and triethyl aluminum (TEA) lead to extremely different products, such as 91% selective 1-butene (I), adjustable Schulz–Flory α-olefin distributions (II) or III olefin distributions (blue) similar to Poisson distributed products of the pure CCTP runs (red, see also Table 1 entries 1 and 2). The corresponding Y–Al–Ni ratios (I–III) responsible for these different products are given in the table

**Table 1 Time-dependent CCTP of ethylene using catalyst 1 and 1 mmol of TEA**

$$n \; \diagup\!\!\!\diagup \xrightarrow[\text{2) acidic workup}]{\text{1) } \mathbf{1}\,/\,\text{AlEt}_3} \quad \diagdown\!\diagup\!\diagdown\!\diagup\!\diagdown\!\diagup\!\diagdown \;_{n-2}$$

| Entry | Time (min) | Ni precatalyst | $V_{eth}$ (L) | $M_n$ (g mol$^{-1}$) | α-Olefin content (mol%) | PDI | Productivity[a] |
|---|---|---|---|---|---|---|---|
| 1 | 5 | – | 1.8 | 450 | 0 | 1.3 | 2,700 |
| 2 | 10 | – | 2.8 | 1,150 | 0 | 1.2 | 2,100 |
| 3 | 15 | – | 4.0 | 1,540 | 0 | 1.2 | 2,000 |
| 4 | 20 | – | 4.7 | 1,770 | 0 | 1.2 | 1,750 |
| 5 | 25 | – | 5.4 | 2,250 | 0 | 1.1 | 1,620 |
| 6 | 30 | – | 5.9 | 2,560 | 0 | 1.1 | 1,400 |
| 7 | 30 | [Ni(acac)$_2$] | 6.2 | n.d. | 80 | n.d. | 1,550 |
| 8 | 30 | [Ni(O$_2$CR$^1$)$_2$] | 3.8 | n.d. | 90 | n.d. | 970 |
| 9 | 30 | [Ni(cod)$_2$] | 4.8 | n.d. | 99 | n.d. | 1,200 |

*n.d.* not determined, R$^1$ = C$_7$H$_{15}$

Reaction conditions: catalyst **1** ($n = 10$ μmol), $p_{eth} = 9.0$ bar, $T = 80$ °C, $n_{TEA} = 1$ mmol, $V_{tol} = 250$ ml

[a]kg$_{ethylene}$ mol$_{cat}^{-1}$ h$^{-1}$

The yttrium catalyst forms alkyl Al compounds via CCTP starting from TEA and ethylene. In parallel, a displacement catalyst, here Ni-based, recycles the TEA via β-hydride elimination/transfer forming LAOs that match the length of the alkyl chains formed. Thus, CTA can be used substoichiometric or formally catalytic. Not all three ethyl groups on aluminum have to be extended; partial chain exchange to Al and extension can be sufficient. Assuming both reactions are not zero order regarding the catalysts, the variation of the catalyst concentration permits the independent tuning of the rates of one or the other partial reaction, CCTP and displacement. If the rate of the CCTP step is increased and/or that of the displacement step is reduced, short chain LOAs can be produced up to the selective 1-butene formation. Inversely, long chain LOA distributions should be accessible. We expected the use of catalysts based on very different transition metals is crucial, for example, a group 3 and a group 10 metal catalyst, to avoid poisoning between the two catalysts. Both metals show a very different coordination chemistry, and ligands that stabilize one metal only bind weakly to the other. Olefins, for instance, bind comparatively weakly to Y[13]. Thus, the stabilization of the displacement catalyst by olefin ligands would not or only weakly harm the active site of the group 3 metal polymerization catalyst if both catalysts are combined. We used a well-defined yttrium catalyst for the CCTP step[14, 15]. Its molecular structure determined by X-ray single crystal structure analysis is shown in Fig. 1b (Supplementary Fig. 1). Anilinium borate [PhNHMe$_2$][B(C$_6$F$_5$)$_4$] was used to activate the corresponding guanidinato-dialkyl precatalyst (Supplementary Fig. 2) complex. See Supplementary Methods for more information of the synthesis of the CCTP catalyst. Notably, the metal center is stabilized by η$^6$-arene coordination by *N,N*-dimethylaniline[16]. Controlled CCTP has been observed in toluene in the presence of ethylene (9.0 bar) and TEA (Al/Y=100/1). Samples taken during a CCTP run indicate an increase in the molecular weight of the polyethylene samples obtained over time. A narrow polydispersity indicating a very fast and reversible chain transfer between the CCTP catalyst and the Al-based CTA is observed under the conditions applied (Table 1). Furthermore, different Ni complexes were additionally added to the CCTP run listed as entry 6 in Table 1. The nickel compounds were added as stock solutions in toluene and the products were analyzed after 30 min via gas chromatography, $^1$H NMR and high temperature gel permeation chromatography.

The best results regarding activity and α-olefin content or selectivity were obtained by using the [Ni(cod)$_2$] (cod=cycloocta-1,5-diene). A comparison of the productivity of the pure CCTP run (Table 1, entry 6) and the tandem process with [Ni(cod)$_2$] as the precatalyst (Table 1, entry 9) revealed no significant catalyst poisoning. No branched or internal olefins were obtained and the linear α-olefins follow a Schulz–Flory distribution[17, 18]. Gas consumption profiles of entry 9 in Table 1 (Supplementary Fig. 11) indicate that the trimetallic catalyst system is still active after 30 min. The half-life of the catalyst system, calculated from this profile assuming a linear decay, is 19 min. We concluded that 30 min is a suitable time for further experiments applying the trimetallic catalyst system. The other Ni-based precatalysts either reduce the activity of the CCTP catalyst or are less selective regarding LAO formation. To find an explanation for the fast and highly selective β-hydride elimination/transfer reaction of aluminum alkyls by [Ni(cod)$_2$], we investigated the conversion of trioctylaluminum into 1-octene and TEA by NMR spectroscopy (Fig. 1c). Trioctylaluminum and [Ni(cod)$_2$] were placed in a Young valve NMR tube followed by the condensation of an excess of ethylene. The reaction was started by heating the mixture to the temperature desired. Different Ni catalyst concentrations were applied and the initial rates were determined. A plot of the initial rates of the Al alkyl displacement reaction vs. the nickel catalyst concentration indicate that the reaction is first order regarding Ni (Fig. 1c). We assume a nickel complex based ß-hydride elimination/transfer process as proposed by Wilke/Eisch and coworkers[19, 20].

**Tunability of the Y–Al–Ni concept**. Ideally, the product(s) formed in a tandem process should be altered independently by changing the concentration of the Y or Ni catalysts. In addition, different product distributions can be synthesized by changing the concentration of ethylene or TEA. Here, a change in concentration may affect both partial reactions. An increase in the TEA concentration, for instance, reduces the rate of chain growth (inverse first order)[9] and increases the rate of the Ni-mediated β-hydride elimination/transfer.

The influence of the concentration of the Y-based CCTP catalyst on the α-value of the α-olefin distribution was examined by applying 5–15 μmol of **1** in tandem runs (Table 2, entries 1–4). As expected, the α-value increases with a higher Y catalyst concentration. A higher Y catalyst concentration leads to faster

**Table 2 Influence of the Y and Ni catalysts concentration as well as TEA concentration and ethylene pressure on the $\alpha$-value, TON Al (formal Al turnover number) and productivity**

$$n \diagup \quad \xrightarrow[\text{2) acidic workup}]{\text{1) } \mathbf{1} \text{ / AlEt}_3 \text{ / Ni(cod)}_2} \quad \diagdown\!\!\diagup\!\!\diagdown\!\!\diagup_{n-2}$$

| Entry | $p_{eth}$ | $n_{Ni}$ ($\mu$mol) | $n_Y$ ($\mu$mol) | $n_{TEA}$ (mmol) | $\alpha$-value | TON Al | Productivity[a] |
|---|---|---|---|---|---|---|---|
| 1 | 9 | 2 | 5 | 1 | 0.48 | 15 | 900 |
| 2 | 9 | 2 | 7 | 1 | 0.52 | 37 | 1,100 |
| 3 | 9 | 2 | 10 | 1 | 0.56 | 48 | 1,200 |
| 4 | 9 | 2 | 15 | 1 | 0.64 | 69 | 1,400 |
| 5 | 9 | 8.0 | 10 | 1 | 0.35 | 30 | 800 |
| 6 | 9 | 4.0 | 10 | 1 | 0.47 | 33 | 960 |
| 7 | 9 | 1.0 | 10 | 1 | 0.70 | 40 | 1,200 |
| 8 | 9 | 0.5 | 10 | 1 | 0.84 | 39 | 1,300 |
| 9 | 9 | 0.2 | 10 | 1 | – | 4 | 1,120 |
| 10 | 9 | 4 | 10 | 7.0 | 0.09 | 5 | 500 |
| 11 | 9 | 2 | 10 | 6.0 | 0.13 | 4 | 550 |
| 12 | 9 | 2 | 10 | 4.0 | 0.21 | 7 | 700 |
| 13 | 9 | 2 | 10 | 2.0 | 0.49 | 15 | 850 |
| 14 | 9 | 2 | 10 | 0.5 | 0.80 | 78 | 1,650 |
| 15 | 4.0 | 2 | 10 | 1 | 0.40 | 15 | 520 |
| 16 | 6.5 | 2 | 10 | 1 | 0.45 | 36 | 850 |
| 17 | 11.5 | 2 | 10 | 1 | 0.63 | 50 | 1,400 |
| 18 | 14.0 | 2 | 10 | 1 | 0.71 | 54 | 1,600 |

Reaction conditions: catalyst **1** ($n = 10$ $\mu$mol), $T = 80$ °C, $t = 30$ min, $V_{tol} = 250$ ml
[a]$\text{kg}_{ethylene}$ $\text{mol}_{cat}^{-1}$ $\text{h}^{-1}$

chain growth and the formation of longer Al alkyl chains prior to the Ni-mediated chain displacement, which proceeds at the same rate for all experiments listed in Table 2, entries 1–4. Interestingly, the productivity (given in $\text{kg}_{ethylene}$ $\text{mol}_{cat}^{-1}$ $\text{h}^{-1}$) also increases with increasing Y catalyst concentration. The catalyst's productivity is also affected by the catalyst-CTA ratio[9]. The state of the active polymerization is in equilibrium with the chain-transfer state where the alkyl chains are transferred via a bimetallic complex towards Al[21–23]. Lower CTA to Y catalyst ratios resulting from a higher Y catalyst concentration will shift the equilibrium towards the chain-growing state leading to an increase in productivity. It is highly interesting that the turnover number (TON) for Al is significantly increased at higher Y catalyst concentrations. The TON of Al is calculated by dividing the moles of the $\alpha$-olefins obtained by the moles of TEA used. One would expect that the rate of the displacement reaction is similar for all runs, since the number of alkyl Al chains and the Ni catalyst concentration are the same. Consequently, the real Al TON should be the same as well. The way we measure the TON of Al shown in Table 2 does not display the real Al TON, since we do not see ethyl recycling. We only monitor the recycling of chains longer than ethyl. The number of these chains increase with increasing Y catalyst concentrations. An increase of the Y catalyst concentration leads to an extension of the Al alkyl chain lengths, indicated by the higher $\alpha$-values, and, in addition, to an increase of the number of extended Al alkyl chains, indicated by the increased TON of Al.

The results (Table 2, entries 5–8, Supplementary Figs. 3 and 4) show an expected dependence of the $\alpha$-value from the Ni catalyst concentration. Higher Ni concentrations lead to a faster displacement of built up and transferred Al alkyl chains. As a result, the higher the Ni concentration, the smaller the $\alpha$-value of the $\alpha$-olefin distribution obtained will be. We believe that there is a maximum amount of Ni concentration in this system. The

maximum is reached if every alkyl chain gets immediately displaced directly after its first transfer from the CCTP catalyst to Al. If chain transfer is significantly faster than chain growth, selective 1-butene formation becomes feasible (vide infra). The smallest chain extension possible is transferred and displaced immediately in such a case. At a very low concentration of the displacement catalyst, not all built up alkyl chains are displaced during the tandem reaction and a fraction of alkanes related to the amount of CTA is formed after the workup. Similarly, a very slow displacement reaction can lead to mixtures of alkanes and olefins even at high displacement catalyst concentrations[11]. Interestingly, the TON of Al is about the same in all experiments listed in Table 2, entries 5–8. One would expect an increase in the TOF of Al with a faster displacement reaction resulting from a higher Ni catalyst concentration. Here again, the number of extended Al alkyl chains is relevant, since we can only monitor their β-hydride elimination/transfer. Because the Y catalyst concentration and the CTA concentration are the same in all experiments listed in entries 5–8 in Table 2, the number of extended chains is similar to the TON of Al. The increase of the productivity can be explained as less blocking of the polymerization catalyst by the displacement catalyst.

We were next interested in how the $\alpha$-value of the $\alpha$-olefins produced is influenced by varying the Al alkyl concentration (see Table 2, entries 11–14, see Supplementary Fig. 5). Different amounts of TEA from 0.5 to 6.0 mmol were added as stock solutions to the tandem reaction. We noticed a strong effect of the TEA concentration on the $\alpha$-value of the $\alpha$-olefins obtained. The $\alpha$-value can be adjusted by increasing the CTA concentration over a large range from 0.80 to 0.13, which are quite different olefin distributions (Supplementary Fig. 6). Changes in the TEA concentration has two major impacts on the tandem reaction. Since the catalyst productivity is inverse first order regarding the Al alkyl concentration, high amounts of Al lead to low monomer

insertion. Consequently, relatively short chains are transferred to Al. Another effect of high Al concentrations is an increase in the chain displacement rate. Chain displacement is first order regarding Al. The constructive combination of both influences permits the alteration of the $\alpha$-value over such a large range.

Finally, we were interested in how the pressure of ethylene influences the product distribution obtained (Table 2, entries 15–18). The solubility of ethylene in toluene is influenced by the pressure applied and, therefore, the rate of monomer insertion can be tuned. We observed the expected increase of the $\alpha$-value with increasing ethylene pressure (Supplementary Figs. 7 and 8). The length of the mean alkyl chain formed at Al is dependent mainly on the number of monomer insertions done by the CCTP catalyst within a given time frame, where time is defined as the mean time after which displacement takes place. It has been kept constant (Table 2, entries 15–18). The productivity increases linearly with increased pressure over the pressure range investigated. The increasing TON of Al can again be explained by the presence of more extended alkyl Al chains in the high-pressure experiments.

With our knowledge about the flexibility of the Y–Al–Ni catalyst system, we can now adjust the catalyst system to obtain many of the product distributions desired (Fig. 1d). This led us to push towards extremes regarding the products formed (Table 2, entries 9 and 10). The CTA concentration was varied, since it has the strongest influence on the $\alpha$-value (Table 2, entry 10). In addition, we varied the Ni concentration. Ethylene pressure and Y catalyst concentration were kept constant. A very small $\alpha$-value of 0.09 can be obtained by using 7.0 mmol CTA and a Y–Ni ratio of 2.5. More than 90% of the product obtained consists of 1-butene (Table 2, entry 10). A second extreme product distribution can be obtained by using 1.0 mmol CTA and very small amounts of the Ni precursor (Y/Ni=50). This experiment leads to olefins with chain lengths centered around 850 gmol$^{-1}$ with a $M_w/M_n$ of 1.3 (HT-GPC) and a vinyl content of 92% ($^1$H NMR, see Supplementary Fig. 9) (Table 2, entry 9). This product distribution (blue, Fig. 1d, III) is similar to Poisson distributed runs of the pure CCTP polymerization (Table 1, entries 1 and 2, red in Fig. 1d, III).

## Discussion

A highly flexible trimetallic synthesis concept of LAOs from ethylene has been introduced. The selective formation of an α-olefin, namely 1-butene, or of various LAO distributions with one catalyst system has been demonstrated. The catalyst system consists of a Y-based CCTP catalyst and an Ni-based ß-hydride elimination/transfer catalyst. Both catalysts do alkyl or polymeryl chain transfer with Al alkyls in a similar rate regime. The Y catalyst permits controlled chain growth at Al, forming trialkylaluminum from ethylene and TEA. The Ni catalyst mediates β-hydride elimination/transfer forming olefins matching the chain length of the alkylaluminum chains grown and transferred by the Y catalyst. In addition, TEA is formed and, thus, recycled. The key to the many product spectra or selective LAO formation is the (independent) adjustability of the rates of the Y- and Ni-catalyzed sub-processes. We have demonstrated that these rates can be adjusted by the concentration of the Y catalyst, the Ni catalyst and the chain-transfer agent, and the ethylene pressure. In addition, it is important that the two catalysts do not poison each other significantly. Our broadly tunable synthesis concept of LAOs is also substoichiometric or formally catalytic regarding Al. The problem of low catalyst economy in CCTP is addressed by multiple usage of the chain-transfer agent.

## Methods

**General**. All manipulations of air sensitive compounds were performed with exclusion of oxygen and moisture using standard Schlenk techniques or a nitrogen or argon filled glove box (mBraun) with a high capacity circulator (<0.1 p.p.m. O$_2$). All solvents used for air and moisture sensitive reactions were dried and purified by distillation from Na/benzophenone or CaH$_2$ (halogenated solvents) under argon atmosphere.

**Typical semi batch poly-/oligomerization run**. The autoclave was evacuated (1×10$^{-3}$ mbar) and heated at 100 °C for 30 min. Afterwards, the polymerization temperature was adjusted, charged with the desired amount toluene (typically 250 ml) and pressurized with ethylene. Subsequently activator, CTA and nickel solution were added as stock solutions via a syringe. Afterwards the catalyst (1 ml) stock solution was added. The ethylene flow was measured over the course of the polymerization procedure and is given as $V_{eth}$ under normal conditions. After the procedure the reaction mixture was quenched with acidified ethanol and the obtained products were analyzed.

**Data availability**. Crystallographic data CCDC 1570423-1570424 contains the Supplementary crystallographic data for this paper. The data can be obtained free of charge from The Cambridge Crystallographic Data Centre via www.ccdc.cam.ac.uk/structures. All other data is available from the authors upon reasonable request

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

## Acknowledgements

We thank Stefan Schwarz for the simulation of the product distributions. We thank SASOLGermany GmbH and the DFG, KE-756/21-2, for the financial support.

## Author contributions

A.G. carried out the synthesis of the catalyst, the poly-/oligomerizations and the kinetic experiments. T.D. carried out the X-ray characterization of the catalyst. W.P.K. carried out the high temperature GPC analysis. A.G. and R.K. designed the experiments and co-wrote the manuscript.

## Additional information

**Competing interests:** The authors declare the following competing financial interest(s): A.G., W.P.K. and R.K. are inventors of patent applications WO2016180539 and WO2016180538, filed: May 13, 2015, published: 16.11.2016. The remaining authors declare no competing financial interests.

