## [Peer Review File · Nature Communications]

Reviewers' comments:

Reviewer #1 (Remarks to the Author):

The manuscript of Kempe and coworkers details the results of a successful development of a catalyst that can convert ethylene into a distribution of linear alpha-olefins (LAOs). Several unique features of this system are presented, including the ability to conduct reversible coordinative chain transfer polymerization (CCTP) in the presence of a nickel co-catalyst which now provides a new pathway for chain-termination that generates a LAO and recycles the trialkylaluminum chain transfer agent. The fact that all three metal components of this system are both compatible and synergistic is impressive and the inclusion of the Ni termination catalyst now provides an additional parameter that can be used to control product distribution in a controlled fashion. Indeed, the authors demonstrate the ability to tune products from almost pure 1-butene to LAOs with Mn values well above those that can be obtained by existing technologies. Since the scientific and commercial ramifications of this breakthrough extend beyond fundamental chemistry, these findings should be of broad interest to the readers of Nature Chemistry. The experimental work has been performed to a very high standard and the manuscript is well written and easy to follow - even for the non-specialist. Congratulations to the authors on achieving a significant and very nice advance. A recommendation to publish 'as-is' is provided.

Reviewer #2 (Remarks to the Author):

This is an interesting report on the development of a new, trimetallic catalyst system of high potential industrial importance that allows the synthesis of alpha-olefins from ethylene, ranging from simple ethylene dimerization to oligomerization to give long-chain olefins with molecular weights of nearly 1000. Unusually, the product distribution can be controlled by adjusting the relative concentration of catalyst (Y), chain transfer agent (TEA) and transfer catalyst (nickel). Similar reactions have been attempted before, but without much success since several kinetic parameters must correspond to one another for the system to work. The authors have hit on such a system.

The work is high quality and clearly described. It is suitable for publication essentially as it stands. There are just a few questions:

(1) What is the half-life of this catalyst system? Reactions of up to 30 min duration are reported. Many highly active catalysts are prone to more or less fast deactivation. On the other hand, soluble lanthanide catalysts are often deactivated e.g. by formation of hydride-bridged species which are comparatively stable.

The usefulness of the catalyst would be greatly increased if it could be shown that this is not the case here. Perhaps gas consumption profiles can be added to the SI. Under suitable conditions, long-lived catalysts may produce much higher TONs than those reported here. Or do the reported figures constitute the maximum?

(2) Obviously detailed mechanistic investigations are beyond the scope of this paper, but one must wonder about the mode of beta-H elimination with Ni(0) as catalyst. Transfer of an alkyl chain from Al to Ni requires that the Ni center has an anionic ligand that can be substituted. At the same time, two alkyl ligands are not transferred to Ni at the same time since this would lead to H-transfer and formation of an alkane, at about the same rate of alkene formation. It would be interesting to hear the authors' view on this aspect. Could the Ni action possibly be heterogeneous, i.e. nanoparticles?

Reviewer #1:

“The manuscript of Kempe and coworkers details the results of a successful development of a catalyst that can convert ethylene into a distribution of linear alpha-olefins (LAOs). Several unique features of this system are presented, including the ability to conduct reversible coordinative chain transfer polymerization (CCTP) in the presence of a nickel co-catalyst which now provides a new pathway for chain-termination that generates a LAO and recycles the trialkylaluminum chain transfer agent. The fact that all three metal components of this system are both compatible and synergistic is impressive and the inclusion of the Ni termination catalyst now provides an additional parameter that can be used to control product distribution in a controlled fashion. Indeed, the authors demonstrate the ability to tune products from almost pure 1-butene to LAOs with Mn values well above those that can be obtained by existing technologies. Since the scientific and commercial ramifications of this breakthrough extend beyond fundamental chemistry, these findings should be of broad interest to the readers of Nature Chemistry. The experimental work has been performed to a very high standard and the manuscript is well written and easy to follow - even for the non-specialist. Congratulations to the authors on achieving a significant and very nice advance. A recommendation to publish 'as-is' is provided.”

“Our response”: Thanks to Reviewer #1 for evaluating our manuscript!

Reviewer #2:

“This is an interesting report on the development of a new, trimetallic catalyst system of high potential industrial importance that allows the synthesis of alpha-olefins from ethylene, ranging from simple ethylene dimerization to oligomerization to give long-chain olefins with molecular weights of nearly 1000. Unusually, the product distribution can be controlled by adjusting the relative concentration of catalyst (Y), chain transfer agent (TEA) and transfer catalyst (nickel).

Similar reactions have been attempted before, but without much success since several kinetic parameters must correspond to one another for the system to work. The authors have hit on such a system.

The work is high quality and clearly described. It is suitable for publication essentially as it stands.”

“Our response”: Thanks to Reviewer #2 for evaluating and improving our manuscript!

“There are just a few questions:

(1) What is the half-life of this catalyst system? Reactions of up to 30 min duration are reported. Many highly active catalysts are prone to more or less fast deactivation. On the other hand, soluble lanthanide catalysts are often deactivated e.g. by formation of hydride-bridged species which are comparatively stable. The usefulness of the catalyst would be greatly increased if it could be shown that this is not the case here. Perhaps gas consumption profiles can be added to the SI. Under suitable conditions, long-lived catalysts may produce much higher TONs than those reported here. Or do the reported figures constitute the maximum? “

“Our response”: Very interesting points! The gas (ethylene) consumption profiles of Entry 6 (Table 1), 30 min run with pure CCTP and Entry 9 of this table [same conditions only with Ni(COD)₂] indicate that the catalyst systems are still active after 30 min since ethylene is still consumed. We used 30 min runs to be in a time range in which the catalyst is still active at the end of the reaction time. The half-life of the trimetallic catalyst system calculated from the gas consumption profile of Entry 9 (Table 1), assuming a linear decay, is 19 min. The reported figures constitute nearly the maximum with regard to TON.

“Our alteration”: We added the two above mentioned gas consumption profiles to the SI and added the following short text section to the manuscript. “Gas consumption profiles of Entry 9 (Table 1) indicate that the trimetallic catalyst system is still active after 30 min. The half-life of the catalyst system, calculated from this profile assuming a linear decay, is 19 min. We concluded that 30 min is a suitable time for further experiments applying the trimetallic catalyst system”

“(2) Obviously detailed mechanistic investigations are beyond the scope of this paper, but one must wonder about the mode of beta-H elimination with Ni(0) as catalyst. Transfer of an alkyl chain from Al to Ni requires that the Ni center has an anionic ligand that can be substituted. At the same time, two alkyl ligands are not transferred to Ni at the same time since this would lead to H-transfer and formation of an alkane, at about the same rate of alkene formation. It would be interesting to hear the authors’ view on this aspect. Could the Ni action possibly be heterogeneous, i.e. nanoparticles?”

“Our response”: Again, very interesting point! We assume a nickel β-hydride elimination/transfer mechanism as suggested by G. Wilke and co-workers [ref #1: Fischer, K., Jonas, K., Misbach, P., Stabba, R. & Wilke, G. Zum “Nickel-Effekt” *Angew. Chem. Int. Ed.* **12**, 943-1026 (1973)]. Ni(II) complexes are reduced to Ni(0) species in the presence of trialkylaluminum compounds and are stabilized by olefin ligands such as ethylene. The Ni(0)-ethylene complexes form adducts with trialkylaluminum, which can mediate an electrocyclic reorganization process to form dialkylethylaluminum and a Ni(0)-alpha-olefin complex. Wilke/Eisch and coworkers [ref. #2: Eisch, J. J., Ma, X., Singh, M. & Wilke G. Aluminum-nickel bonded intermediates in the Ziegler Nickel Effect: mechanistic support from catalyzed hydroalumination and carboalumination reactions *J. Organomet. Chem.* **527**, 301-304 (1997)] later specified that Ni-olefin complexes can insert into the Al-carbon bond of trialkylaluminum compounds. They could demonstrate (NMR) that Ni(COD)₂ can insert into an Al-carbon bond of trialkylaluminum compounds forming (COD)Ni(alkyl)-Al(alkyl)₂. Now β-hydride elimination/transfer to monomer (ethylene) becomes feasible forming (COD)Ni(ethyl)-Al(alkyl)₂ and the corresponding alpha-olefin. Of course, nanoparticle formation cannot be ruled out completely. It, most likely, becomes especially relevant at low ethylene pressures.

“Our alteration”: We added the following sentence to the manuscript: “We assume a nickel complex based β-hydride elimination/transfer process as proposed by Wilke/Eisch and coworkers. Ref # 1 and 2”

REVIEWERS' COMMENTS:

Reviewer #2 (Remarks to the Author):

The authors have satisfactorily addressed all the points raised. The paper is recommended for publication.